# Rapid Internalization and Nuclear Translocation of CCL5 and CXCL4 in Endothelial Cells

**DOI:** 10.3390/ijms22147332

**Published:** 2021-07-08

**Authors:** Annemiek Dickhout, Dawid M. Kaczor, Alexandra C. A. Heinzmann, Sanne L. N. Brouns, Johan W. M. Heemskerk, Marc A. M. J. van Zandvoort, Rory R. Koenen

**Affiliations:** 1Department of Biochemistry, Cardiovascular Research Institute Maastricht, Maastricht University, 6229 ER Maastricht, The Netherlands; a.dickhout@maastrichtuniversity.nl (A.D.); d.kaczor@maastrichtuniversity.nl (D.M.K.); a.heinzmann@maastrichtuniversity.nl (A.C.A.H.); sanne.brouns@maastrichtuniversity.nl (S.L.N.B.); jwm.heemskerk@maastrichtuniversity.nl (J.W.M.H.); 2Department of Genetics and Cell Biology, Molecular Cell Biology, School for Oncology and Developmental Biology, Maastricht University, 6229 ER Maastricht, The Netherlands; mamj.vanzandvoort@maastrichtuniversity.nl; 3Institute for Molecular Cardiovascular Research IMCAR, RWTH Aachen University, 52074 Aachen, Germany; 4Institute for Cardiovascular Prevention (IPEK), LMU Munich, 80336 Munich, Germany

**Keywords:** RANTES, platelet factor 4, chemokine, endothelial cell, monocyte

## Abstract

The chemokines CCL5 and CXCL4 are deposited by platelets onto endothelial cells, inducing monocyte arrest. Here, the fate of CCL5 and CXCL4 after endothelial deposition was investigated. Human umbilical vein endothelial cells (HUVECs) and EA.hy926 cells were incubated with CCL5 or CXCL4 for up to 120 min, and chemokine uptake was analyzed by microscopy and by ELISA. Intracellular calcium signaling was visualized upon chemokine treatment, and monocyte arrest was evaluated under laminar flow. Whereas CXCL4 remained partly on the cell surface, all of the CCL5 was internalized into endothelial cells. Endocytosis of CCL5 and CXCL4 was shown as a rapid and active process that primarily depended on dynamin, clathrin, and G protein-coupled receptors (GPCRs), but not on surface proteoglycans. Intracellular calcium signals were increased after chemokine treatment. Confocal microscopy and ELISA measurements in cell organelle fractions indicated that both chemokines accumulated in the nucleus. Internalization did not affect leukocyte arrest, as pretreatment of chemokines and subsequent washing did not alter monocyte adhesion to endothelial cells. Endothelial cells rapidly and actively internalize CCL5 and CXCL4 by clathrin and dynamin-dependent endocytosis, where the chemokines appear to be directed to the nucleus. These findings expand our knowledge of how chemokines attract leukocytes to sites of inflammation.

## 1. Introduction

Chemokines are small chemotactic cytokines that have an important role in regulating leukocyte trafficking during health and disease [1,2]. Through binding and activation of their cognate G protein-coupled receptors, they can rapidly induce leukocyte responses e.g., integrin activation, flow-resistant arrest, cell polarization, and transendothelial migration to sites of inflammation or infection. On a structural level, chemokines are hallmarked by a disordered N-terminus, a 3-strand antiparallel β-sheet, and a C-terminal α-helix. In addition, stretches of basic amino acids mediate binding to glycosaminoglycans (GAGs), e.g., heparin, heparan sulfate, and similar sulfated polysaccharides that constitute the cellular glycocalyx [3,4]. This warrants immobilization of the chemokines to the cell surface, e.g., of endothelial cells (EC) of the vessel wall, allowing them to be visible by rolling leukocytes. Besides this concept of chemokine presentation on the endothelial surface, constituting a message for leukocytes, some chemokines might be produced by the EC themselves and stored in small vesicles below the apical cell surface, which can be located by adherent monocytes prior to diapedesis [5]. In addition, chemokines on the vessel wall might originate from the subendothelial tissue and move to the vascular surface by transcytosis [6,7], yet they can also be deposited on the vessel wall by rolling platelets, as was shown for CCL5 (RANTES) [8]. This chemokine transfer to EC by activated platelets was shown to facilitate subsequent monocyte arrest [8,9]. Infusion of activated platelets into hyperlipidemic mice resulted in an accelerated development of atherosclerosis, which could be attributed in part to increased immobilization of CCL5 onto the atherosclerotic vessels [10]. Interestingly, CCL5 and CXCL4 (platelet factor 4), one of the most abundant chemokines in platelets, can interact with each other to form heterodimers, which are particularly potent in the recruitment of monocytes [11] and were shown to modulate the severity of atherosclerosis, stroke, abdominal aneurysm, and myocardial infarction in mice [12,13,14,15,16]. Although the interaction of CCL5 with GAGs has been postulated as essential for function in vitro and in vivo [17], the exact mechanism of CCL5 presentation to the cell surface and recognition by immune cells is incompletely characterized. Although the presence of CXCL4 led to increased binding of CCL5 to the surface of monocytic cells, it is unclear whether this explains the synergy between those chemokines [11]. A previous study has indicated that CCL5 is immobilized on the surface of human umbilical vein endothelial cells (HUVEC) in filamentous flow-resistant polymers, which might form a scaffold for leukocyte recruitment [18]. Interestingly, part of the CCL5 was observed intracellularly. To elaborate on the previous findings and to further investigate the mechanisms that underlie chemokine-induced leukocyte recruitment, we investigated the fate of exogenously added CCL5 and CXCL4 to EC. We found that incubation of EC with CCL5 and CXCL4 under static conditions led to rapid internalization of the chemokines, where CXCL4 remained partly presented on the cell surface. Internalization was an active process and dependent on G protein-coupled receptor (GPCR) signaling and classic endocytosis and resulted in calcium signaling within endothelial cells. Remarkably, internalized CCL5 and CXCL4 were targeted to the nucleus. Leukocyte arrest was not altered upon pretreatment with chemokines.

## 2. Results

### 2.1. Surface Presentation of the Chemokines CCL5 and CXCL4 on EC

To initially investigate the interaction of chemokines with endothelial cells, cells from the line EA.hy926 (EAHy) were incubated without or with CCL5 and CXCL4, for a prolonged time of 60 min at 37 °C. The cells were subsequently stained using specific fluorescent antibodies without prior permeabilization. Thus, only the extracellular fraction of CCL5 and CXCL4 would be visible. Absence of exogenous chemokines before staining did not result in a notable fluorescent signal for either CCL5 or CXCL4 (Figure 1A,F). Likewise, the fluorescent intensity did not notably increase after 60 min treatment of EAHy cells with CCL5 (Figure 1B). However, incubation of EAHy with CXCL4 led to a robust fluorescent signal (Figure 1G). Washing the EAHy cells with heparin after incubation with chemokines, but prior to antibody staining, led to loss of fluorescent signal (Figure 1C,H). Co-staining of confluent EAHy cells with CD31 and CCL5 or CXCL4, respectively, revealed a cytoplasmic staining pattern of the chemokines that was distinct from the typical accumulation of the CD31 signal at the cell-cell contacts (Figure 1D,E,I,J). Imaging along the *z*-axis implied faint CCL5 at the luminal aspect of the EAHy cells and increased staining intensities toward the basolateral side (Appendix A).

### 2.2. Permeabilization of EAHy Increases the CCL5 and CXCL4 Antigen Signal

Because chemokines are known to be retained by EC, the EAHy cells were permeabilized in order to investigate an intracellular presence. After permeabilization and addition of the fluorescent antibodies, minimal staining of CCL5 and CXCL4 was observed in the absence of exogenous chemokines (Figure 2A). This signal might reflect low levels of endogenous CCL5 or CXCL4 (or variant CXCL4L1 [19]) present in EAHy. Interestingly, incubation of EAHy with exogenously added CCL5 and CXCL4 at 37 °C for 60 min, followed by permeabilization and staining, resulted in a high signal intensity of the respective chemokine (Figure 2B). Incubation of EAHy with chemokines at 4 °C did not lead to an increase in fluorescent signal (Figure 2C), suggesting that the intracellular accumulation of CCL5 and CXCL4 is an active and energy-requiring cellular process.

### 2.3. Intracellular Accumulation of CCL5 and CXCL4 Is Time-Dependent

In further experiments, the uptake of CCL5 and CXCL4 was followed in time. The chemokines were added and remained present at various increasing time points at 37 °C. Then, surface-bound and excess chemokines were removed by washing with heparin, and cells were fixed, permeabilized, and stained with specific fluorescent-labeled antibodies. Subsequently, the presence of the chemokines was visualized using fluorescent microscopy. In addition, the chemokine-treated cells were lysed after washing, and intracellular chemokine concentrations were measured by ELISA.

Both CCL5 and CXCL4 appeared to be taken up in a time-dependent manner (Figure 3). An increase in subcellular fluorescent signal was already observed after 5 min of incubation and increased over the 120-min duration of the experiment (Figure 3A,C). This was paralleled by an increase of intracellular CCL5 and CXCL4 antigen as observed with ELISA, with an apparent maximal uptake of CXCL4 after 60 min (Figure 3B,D). This supported the idea that EAHy cells actively and rapidly take up the chemokines CCL5 and CXCL4. For clarity, images with separate color channels are shown in Appendix A. Bovine chemokines from FCS did not cross-react with either antibodies used for fluorescence or ELISA (data not shown).

### 2.4. Internalization of CCL5 and CXCL4 Is Dependent on Dynamin- and Clathrin-Mediated Endocytosis

In order to investigate the manner of chemokine uptake, the EAHy cells were pre-treated with inhibitors of clathrin- (Pitstop2) or dynamin-mediated endocytosis (Dynasore) for 15 min, after which the cells were incubated with the chemokines in the presence of the inhibitors. Then, the same procedure for fixation, staining, and imaging as described above was followed. Of note, the pre-treatment with endocytosis inhibitors led to some detachment of the EAHy cells and loss of monolayer properties. Both inhibitors abolished the intracellular uptake of CCL5 and CXCL4 (Figure 4A–H). In addition, blockade of chemokine receptor- and other GPCR-induced signaling by pertussis toxin also led to a reduced internal presence of CCL5 and CXCL4. Effective inhibition of clathrin- and dynamin-dependent endocytosis by the above inhibitors was demonstrated by measurement of DiI-labeled nLDL uptake (Figure 4I). Surprisingly, enzymatic removal of GAGs from the cell surface resulted in an increased uptake of both CCL5 and CXCL4 into EAHy cells (Appendix A). Although chemokine uptake was abolished by pertussis toxin, antagonists of the CCL5 receptor and of the putative CXCL4 receptor CXCR3 prevented neither CCL5 nor CXCL4 internalization (Appendix A). Interestingly, pre-incubation of EAHy cells with CXCL4 prior to the addition of CCL5 led to a significant reduction of subsequent CCL5 uptake (Figure 5A). By contrast, pre-incubation of the cells with CCL5 prior to CXCL4 addition did not affect CXCL4 uptake. This indicates that CXCL4 can desensitize EAHy cells for the uptake of CCL5 (Figure 5B).

To investigate whether EC showed evidence of intracellular signaling upon treatment with CCL5 or CXCL4, HUVECs were loaded with a calcium-sensitive dye and treated with buffer, thrombin as a positive control, or chemokines (Figure 6A–D). Upon stimulation with CCL5 and CXCL4, the HUVECs showed a notable and lasting increase in intracellular calcium levels, which was comparable to that induced by thrombin, as evidenced by classification of the individual cellular calcium mobilization profiles (Figure 6E).

### 2.5. CCL5 and CXCL4 Are Targeted to the Nucleus after Endothelial Uptake

Because uptake of chemokines appeared to be an active process accompanied by cell signaling events, the intracellular fate of CCL5 and CXCL4 was investigated further. Confocal microscopy indicated that the chemokines accumulated in the nucleus 60 min after addition (Figure 7A). This is supported by a high optical resolution Z-stack movie that suggests accumulation of CXCL4 in the nucleoli (Appendix A). In order to further elucidate the subcellular localization of CCL5 and CXCL4, EAHy were lysed after 30 and 120 min of treatment. Subcellular fractions were isolated and measured. Interestingly, both CCL5 and CXCL4 showed an association with the cytoskeleton after 30 min, with a slight further increase at 120 min. However, the nuclear content was only barely increased after 30 min, but strongly increased after 120 min (Figure 7B,C). These data suggest that both CCL5 and CXCL4 are transported to the nucleus through cytoskeletal associations. To investigate whether inflammatory conditions could affect the uptake of chemokines, EAHy were activated using TNFa for 4 or 18 h, prior to addition of CCL5 or CXCL4. However, no difference in chemokine uptake was observed compared with resting cells (Appendix A).

### 2.6. CCL5 and CXCL4 Internalization Does Not Affect Leukocyte Arrest

Previous findings suggested that chemokines e.g., CCL2, stored in subluminal vesicles, could guide lymphocyte tracking on EC [5]. In order to investigate whether a similar mechanism could regulate CCL5- and CXCL4-induced monocyte arrest, HUVECs were incubated with these chemokines for 120 min, treated without or with heparin to remove residual surface chemokines, and subsequently perfused with monocytic MonoMac6 cells. Addition of chemokines for 120 min did not affect MonoMac6 adhesion to HUVECs (Figure 8), indicating that internalized chemokines neither make HUVECs competent for leukocyte interactions nor directly induce leukocyte arrest. Similar results were observed after activation of the HUVECs with TNFa for 4 h, prior to the addition of chemokines (Figure 8). The combined addition of CCL5 and CXCL4, which is a potent stimulus for monocyte arrest, did not increase MonoMac6 adhesion, which indicates that TNFa activation might already support maximal monocyte arrest. These results suggest that at least CCL5 and CXCL4 require surface presentation for leukocyte recruitment.

## 3. Discussion

In this study, we observed that chemokines are taken up by EC in an active manner because uptake did not occur at 4 °C. At 37 °C, intracellular chemokines were detectable as soon as 10 min after addition, and internalized levels approached a steady state after 60 min. This indicates that internalization of the chemokines CCL5 and CXCL4 is a rapid and actively triggered process. Co-staining with CD31 indicated that the chemokines did not co-localize with the cell-cell contacts, and z-stack imaging suggested intracellular rather than luminal localization. The active character of chemokine uptake is further supported by the observed increase of intracellular calcium upon chemokine addition and the blockade of chemokine uptake by the G protein inhibitor pertussis toxin. Together with the observation that chemokine internalization was blocked by inhibitors of clathrin-mediated endocytosis, these findings imply that specific (G protein-coupled) receptors for the uptake of chemokines are involved. However, blockade of CCR5, a receptor for CCL5, by small molecular antagonists did not influence CCL5 uptake, implying that CCR5 does not play a role. In addition, enzymatic removal of GAGs from EC even increased the uptake of CCL5 and did not affect the uptake of CXCL4. Thus, these findings suggest either that binding to GAGs is not important for the uptake of chemokines, that the GAGs involved are not targeted by the enzymatic treatment, or that the remainder of the glycocalyx is sufficient to absorb the chemokines prior to their uptake [3]. Given the highly positive charge of both CCL5 and CXCL4 at physiologic pH, there might be additional molecules that mediate binding of chemokines and support uptake through electrostatic interactions, e.g., negatively charged phospholipids or membrane proteins. In addition, because the endothelial cells were cultured under static conditions in this study, it cannot be excluded that flow-dependent adaptations of the glycocalyx [20] support chemokine binding in a physiologic setting involving blood flow. The absence of flow can be considered a limitation of this study, as well as the 2-dimensional culture conditions under which the endothelial cells were grown. The development of innovative blood vessels-on-a-chip, e.g., to study blood–brain barrier physiology [21], has opened the possibility to investigate chemokine uptake and subsequent monocyte arrest in a system of continuous flow. In addition, molds and tissue supports now enable the culturing of cells in 3 dimensions, which is a further approximation toward physiology [22,23]. Future studies will take these aspects into account, as well as the possible influence of membrane water flux in endothelial cells on chemokine uptake, as can be measured by a recently established calcein quenching-based method [24].

In EC, an alternatively spliced variant of CXCR3, termed CXCR3B, was suggested to serve as a receptor for CXCL4 [25]. However, whether CXCR3 acts as a bona fide receptor for CXCL4 and other angiostatic chemokines is still unclear [26]. In addition, the expression of CXCR3 in EC depends on the cell cycle [27], and CXCR3B signaling is not inhibited by pertussis toxin [25]. Thus, these findings and our observations in this study speak against the involvement of CXCR3B in the internalization of CXCL4. Alternatively, galectins might be involved in chemokine uptake into EC. Galectins are a family of b-galactoside-binding lectins that are involved in various physiologic processes and are established mediators of endocytosis of plasma membrane proteins and glycolipids [28,29]. Recent studies by us and others have shown an interplay between galectins and chemokines [30,31]. For example, there is an interplay between galectin-1 and CXCL4 in platelet activation [30], and CCL5 was shown to undergo physical interactions with galectin-3 [31]. However, whether galectins play a role in the uptake of CCL5 and CXCL4 remains a subject for future investigations.

A somewhat surprising observation is that pre-incubation with CXCL4 can apparently inhibit subsequent CCL5 internalization, but not vice versa. These results show that there is an interdependency between the uptake of CXCL4 and CCL5. We do not have a conclusive mechanistic explanation, but we speculate that a potent internalization induced by CXCL4 may exhaust the uptake machinery for CCL5, or that a CCL5 receptor is co-internalized along with CXCL4, making it less available for CCL5. Future studies are needed to further unravel the mechanisms of chemokine uptake by endothelial cells.

Interestingly, CCL5 was no longer detectable on the surface of EC 60 min after its addition. This appears to be somewhat at odds with the common notion that chemokine presentation onto the luminal surface of the vessel wall directs leukocyte recruitment [32,33]. For CCL5 in particular, binding to GAGs is essential for its leukocyte-recruiting functions [17], and a GAG-binding mutant of CCL5 was shown to act as a dominant-negative antagonist [34]. However, other studies have demonstrated that subluminal presentation of chemokines also can affect leukocyte trafficking. After stimulation with cytokines, the chemokines CCL2 and CXCL10 are stored in vesicles underneath the endothelial membrane in HUVECs [5]. Being localized below the membrane associated with actin, these chemokines are no longer accessible to blocking antibodies yet are still able to trigger the transmigration of lymphocytes by localized release in the tight immunologic synapse [5]. This possibility was also investigated by determining monocytic cell arrest, a well-established function of CCL5, on endothelial cells after incubation with chemokines, without or with removal of residual surface chemokine. After 60 min, no CCL5 was detectable on the surface of EC, implying complete uptake. Possible residual chemokine was removed. Under these conditions, our findings suggest that internalized CCL5 does not induce arrest of MonoMac6 cells on EC, both under resting conditions and after cytokine stimulation. In addition, neither CXCL4 nor CCL5 increase the adhesiveness of EC for monocytic cells under resting or inflammatory conditions.

Internalization of chemokines has been observed in several studies and is considered a physiologic mechanism for their transport from inflamed tissues toward the endothelial lining of the vasculature [6,7]. In polarized Madin–Darby canine kidney cells, rapid internalization within 30 min of the chemokine CCL2 was demonstrated, and CCL2 was transported from the basolateral side to the apical side within 120 min [7]. The atypical chemokine receptor ACKR1, which is also known as Duffy antigen receptor for chemokines (DARC), was shown to mediate the cellular transport of CCL2. Because ACKR1 can both bind CCL5 and CXCL4 [35,36], an involvement of this atypical receptor appears plausible. A previous study has indeed demonstrated an involvement of ACKR1 in the uptake of CXCL1, which is structurally related to CXCL4, in an immortalized HUVEC cell line, yet only when these cells were stably transfected with ACKR1 [37]. Interestingly, the endothelial internalization of CXCL1 was found not to depend on clathrin and only partially on dynamin. Thus, although ACKR1 appears to be a plausible candidate receptor for CCL5 and CXCL4 uptake into EC, it is unclear whether basal expression levels support internalization.

An interesting observation is the nuclear accumulation of the chemokines after uptake into EC. Previous studies have observed a nuclear targeting of the chemokine CXCL12 gamma [38], a splice variant of CXCL12 with a nuclear translocation signal in its C-terminus, and an alternatively spliced form of CCL27, which is targeted to the nucleus by an endogenous nuclear targeting signal [39]. Interestingly, in the latter study by Gortz et al., CCL5 was not targeted to nucleus when expressed in Chinese hamster ovary cells, indicating an absence of an endogenous nuclear translocation signal. Apart from the different cell type, the mechanisms of nuclear targeting might be different in the experimental setting of the two studies and ours. Endogenous expression from a cDNA might sort the newly produced chemokine (precursors) in the endoplasmic reticulum for secretion or for transport to the nucleus, whereas exogenously added chemokine enters the cell through an endocytic pathway. Of note, the association of CCL5 and CXCL4 with the cytoskeletal fractions is indicative of a transport mechanism toward the nucleus. Association of chemokine-loaded vesicles with actin has been observed before [5], yet our experiments yielded no evidence of a packing of CCL5 or CXCL4 into vesicles. Within the nucleus, CXCL4 appeared to accumulate in the nucleolus. The significance of these findings is currently unknown, but CXCL4 is known to have a high affinity for nucleic acids [40], and CXCL4/DNA complexes were recently shown to serve as a ligand for toll-like receptor 9 in the inflammatory activation of dendritic cells [41]. It is tempting to speculate on an active involvement of CXCL4 in transcriptional processes. In monocytes/macrophages, CXCL4 is able to induce considerable changes in the transcriptional landscape, leading to pro-inflammatory and pro-fibrotic phenotypes [42,43]. It is worth noting that unlike other chemokines, no single receptor has been identified that explains all CXCL4 functions. Other cytokine (-like) or danger-associated molecules have intra- and extracellular actions (reviewed in [44]), and it is conceivable that CXCL4 might act in a similar fashion. Alternatively, chemokine uptake by EC might serve to maintain or regulate local chemokine levels in a fashion similar to the ACKRs [44,45] and might be targeted to the nucleus for proteasomal degradation [46].

In conclusion, the results in this study suggest an active and directed uptake of the chemokines CCL5 and CXCL4, resulting in an accumulation toward the nucleus. Although the (patho)physiologic relevance needs to be further characterized, these findings add a further dimension to the cell regulating activities of the chemokine system in health and disease.

## 4. Materials and Methods

### 4.1. Cells

EA.hy926 cells (“EAHy”, ATCC^®^ CRL-292) were cultured in Dulbecco modified Eagle medium (DMEM; Thermo Fisher Scientific, Waltham, MA, USA) supplemented with 10% fetal bovine serum (FBS), 2 mmol/L L-glutamine, 1× HAT (5 mmol/L sodium hypoxanthine, 20 µM aminopterin, and 0.8 mmol/L thymidine) and 100 U/mL penicillin-streptomycin (Thermo Fisher Scientific). Cells were used between passage 7–25. Human Umbilical Vein Endothelial Cells (HUVEC) (Promocell GmbH) were grown in endothelial cell growth medium (Promocell, Heidelberg, Germany) supplemented with 100 U/mL penicillin-streptomycin. Cells were used between passage 4 and 7. MonoMac-6 cells (DSMZ, ACC124) were cultured in RPMI supplemented with 10% FBS, 2 mmol/L L-glutamine, non-essential amino acids, 1 mmol/L sodium pyruvate, and 10 μg/mL human insulin. Cells were used between passage 7 and 25. Bimonthly samples were measured for the absence of mycoplasma using the MycoAlert Mycoplasma Detection Kit as per manufacturer’s protocol (Lonza, Walkersville, MD, USA). All cells were maintained in a humidified incubator at 37 °C and 5% CO_2_.

### 4.2. Chemokine Internalization

Cells were cultured on 8-well Falcon Chambered Cell Culture slides or in 6-well cell culture plates, and incubated with 500 ng/mL recombinant human CCL5/RANTES (R&D Systems, Minneapolis, MN, USA) or recombinant human CXCL4/PF4 (Peprotech, Rocky Hill, NJ, USA), at 4 °C or 37 °C for indicated times, specific per experiment. The chemokine concentration of 500 ng/mL (63 nmol/L was found optimal for triggering monocyte arrest on endothelial cells in previous studies [9,11,14,15]. Native low-density lipoprotein (nLDL) labeled with 1,1′-Dioctadecyl-3,3,3′,3′-Tetramethylindocarbocyanine (Dil-nLDL, 5 µg/mL-Sigma SAE0053) was used as a positive control and to assess the functionality of the endocytosis inhibitors.

In some experiments, it was investigated whether pre-incubation with chemokines could influence internalization. Before the experiment, all culture plates were coated with Attachment Factor (AF, from Gibco). Cells were seeded in a black 96-well plate at a density of 15,000 cells/well and incubated in DMEM, supplemented with 10% FCS for 24 h. The next day, the cells were starved in DMEM supplemented with 0.5% FCS overnight. The following day, the cells were pre-treated with vehicle or 500 ng/mL of the first chemokine (CXCL4 or CCL5) for 30 min. Then, fresh medium (DMEM + 0.5% FCS) containing 500 ng/mL of the other chemokine (CXCL4 or CCL5) was added to the cells, and the culture plates were incubated for an additional 2 h at 37 °C. Then, the cells were incubated with 200 U/mL heparin for 5 min, in order to remove the remaining chemokines from the cell surface. Subsequently, cells were fixed with 4% paraformaldehyde for 10 min and blocked with a blocking buffer (PBS containing 2% BSA and 0.1% Triton X-100) for 1 h at room temperature. Then, the primary antibody (rabbit anti-human PF4 from Peprotech or rabbit anti-human CCL5 from Abcam) was added at a final concentration of 2 µg/mL and incubated overnight at 4 °C. After washing, the secondary antibody was added (goat anti-rabbit, conjugated with AF532 from Life Technologies) at a final concentration of 5 µg/mL for 1 h. Next, cells were washed and nuclei were stained with Hoechst solution. Then, the cell count and the fluorescence were analyzed using a Cytation^TM^ imager. In negative control wells, the addition of primary antibody was omitted.

### 4.3. Inhibitors

Cells were incubated with Dynasore (324410 Merck, Darmstadt, Germany), PitStop2 (ab120687, Abcam, Cambridge, UK), Bordetella pertussis toxin blocking G proteins (Ga_i_, Ga_o,_ and Ga_t_, BML-G101-0050 Enzo Life Sciences, Farmingdale, NY, USA), and DAPTA (2423 R&D Systems). TAK-779 (SML0911), blocking anti-CXCR3 [25] (clone 49801, R&D systems), heparinase III from Flavobacterium heparinum (H8891), chondroitinase ABC from Proteus vulgaris (C2905), and hyaluronidase Type VI-S (H3631) were all obtained from Merck. Neuraminidase (C. perfringens) (P5289) was from Abnova (Taipei City, Taiwan).

### 4.4. Localization

Cells were incubated with recombinant human CCL5/RANTES or recombinant human PF-4/CXCL4, as described above, and fractionated using the subcellular protein fractionation kit for cultured cells (78840-Thermo) as per manufacturer’s protocol. Chemokine quantification in the different subcellular fractions was performed using chemokine-specific sandwich ELISA.

### 4.5. Immunocytochemistry

EA.hy926 cells were cultured on 8-well Falcon Chambered Cell Culture slides (Thermo) and fixed and permeabilized using BD Cytofix/Cytoperm™ as per manufacturer’s instructions. Cells were blocked for 30 min at room temperature using PBS supplemented with 5% FCS. CCL5 was detected using a rabbit polyclonal antibody against RANTES (Abcam ab9679), and CXCL4 was detected using a rabbit polyclonal antibody against platelet factor 4 (Peprotech) at 2 μg/mL. CXCR3 was detected using a monoclonal mouse anti-CXCR3 (R&D systems) at 10 µg/mL. Visualization was performed using donkey-anti-rabbit Alexa Fluor 647 (Thermo) at 5 μg/mL, or goat-anti-mouse FITC (Jackson) at 6 µg/mL. F-actin was visualized using phalloidin Alexa Fluor 488 (Thermo). Antibodies were diluted in PBS supplemented with 5% FCS. Finally, cells were mounted using Vectashield mounting medium containing 4′,6-diamidino-2-phenylindole (DAPI; Vector Laboratories, Burlingame, CA, USA), or stained with Hoechst 33342 (Thermo) and mounted with a glycerol-based mounting medium containing Mowiol 4-88 (Merck). Cells were then imaged using an EVOS FL Cell imaging system, using an Olympus 60× oil objective (1.42NA) and standard filter cubes for DAPI (ex/em 357/447 nm), GFP (ex/em 470/525 nm), and Cy5 (ex/em 628/692 nm). Image overlays and cross-sections were made using Fiji V1.52k [47]. Fluorescence quantification per cell count was analyzed with a Cytation 3 Cell Imaging Multi-Mode Reader (Biotek Instruments Inc., Winooski, VT, USA) using donkey-anti-rabbit Alexa Fluor 532 (Thermo) and Hoechst 33342.

### 4.6. Live Cell Imaging

EA.hy926 cells were grown on 8-well Falcon Chambered Cell Culture slides (Thermo) and incubated for 60 min with CCL5 or CXCL4. Briefly, cells were washed with PBS alone or PBS with 1 mg/mL Heparin to wash away membrane-bound chemokines. Cells were then stained with the respective primary antibodies, and Alexa Fluor 647-coupled secondary antibody and nuclei were visualized with Hoechst 33342 as described before, at 37 °C and 5% CO_2_. Cells were then imaged using an EVOS FL Cell imaging system, using the 20× objective (0.45 NA) and standard filter cubes for DAPI (ex/em 357/447 nm) and Cy5 (ex/em 628/692 nm). Image overlays and cross-sections were made using Fiji V1.52k [47]. All antibodies used against CCL5 and CXCL4 did not detect the bovine chemokine orthologs.

### 4.7. Confocal Imaging

EAhy cells were fixed and stained for CCL5 and CXCL4 as described above. Visualization was performed using Goat-anti-Rabbit FITC (Thermo). Cells were imaged on a Leica TCS SP8 Confocal microscope with a 100× oil immersion/1.4 NA objective and 2× optical zoom, with excitation at 405 or 488 nm, and emissions were collected at 413–480 nm and 498–580 nm, respectively. Images of 512 × 512 pixels were obtained with a pixel size of 0.09 µm, standard pinhole size, and scan speed of 400 Hz. Z-stacks were made with a step size of 0.219 µm (Appendix A) or 0.3 µm (Appendix A).

### 4.8. Calcium Influx

For the influence of chemokines on intracellular calcium concentrations [Ca^2+^]i, HUVECs were cultured in 8-well glass-bottomed ibidi culture slides, coated with 30 µg/mL collagen. After reaching confluency, cells were incubated with 8 µmol/L Fluo-4 acetoxymethyl ester in the presence of 0.4 mg/mL pluronic for 40 min at 37 °C and 5% CO_2_. Changes in [Ca^2+^]i were recorded for 10 min, using a Zeiss LSM 510 confocal microscope (488 nm excitation). At the start of the recording, PBS, chemokines (500 ng/mL each), or thrombin (10 nmol/L) were added to the wells. Changes in fluorescence intensity indicated a spike in cytosolic [Ca^2+^]i and were represented as false-color images (blue: low, green: high). Fluorescence images were analyzed with ImageJ/Fiji software. [Ca^2+^]i spikes were classified based on the type of oscillatory signal in cytosolic-free calcium concentration. Score 1 indicated no to minimal rise of [Ca^2+^]i, Score 2 was indicative of short and low amplitude rises of [Ca^2+^]i, a score of 3 was a single, high rise of [Ca^2+^]i, and a score of 4 was a repetitive and high rise of [Ca^2+^]i.

### 4.9. Quantification of Internalized Chemokines

EA.hy926 cells were cultured in 6-well cell culture plates (Corning, Glendale, AZ, USA), treated with chemokines and/or inhibitors according to the experiment, and washed with 1 mg/mL heparin (180 U/mL) to remove membrane-bound chemokines prior to cell lysis using 400 μL CytoBuster protein extraction reagent (71009-Merck). Cell lysates were centrifuged for 5 min at 16.000× *g* and stored at −20 °C for ELISA. Sandwich ELISA was performed using human CCL5/RANTES (DY278) or human CXCL4/PF4 (DY795) DuoSet ELISA protocols (R&D Systems), respectively, as described [30,48]. Both ELISAs were found to detect neither CCL5 nor CXCL4 in samples of fetal bovine serum.

### 4.10. Laminar Flow-Based Leukocyte Adhesion Assay

Laminar flow-based leukocyte adhesion assay was performed as described in detail previously [49]. Briefly, HUVEC cells were cultured in 35 mm TC-treated cell culture dishes (Thermo) with a density of 1 × 10^5^ cells/cm^2^ for 48 h before stimulating with TNFα (10 ng/mL) for 4 h. Chemokines CCL5 or CXCL4 (both 500 ng/mL) were added for 1 h. MonoMac6 cells were stained with Syto 13 (Thermo) for 30 min at 37 °C and washed and perfused in Hank’s buffer pH 7.45 containing 10 mmol/L Hepes, 3 mmol/L CaCl_2_, 2 mmol/L MgCl_2,_ and 0.2% human serum albumin for 3–6 min at 3 dynes/cm^2^. Adherent cells were counted in 6 view fields and expressed in cells/mm^2^.

### 4.11. Statistical Analysis

Data are presented as mean ± SD unless stated otherwise. Statistical analysis was performed using Graphpad Prism 9 (San Diego, CA, USA), using one-way analysis of variance ANOVA and Sidak’s (parametric) or Dunn’s (non-parametric) post hoc analysis, as indicated. Differences were considered statistically significant at *p* < 0.05 (*).

## Figures and Tables

**Figure 1 ijms-22-07332-f001:**
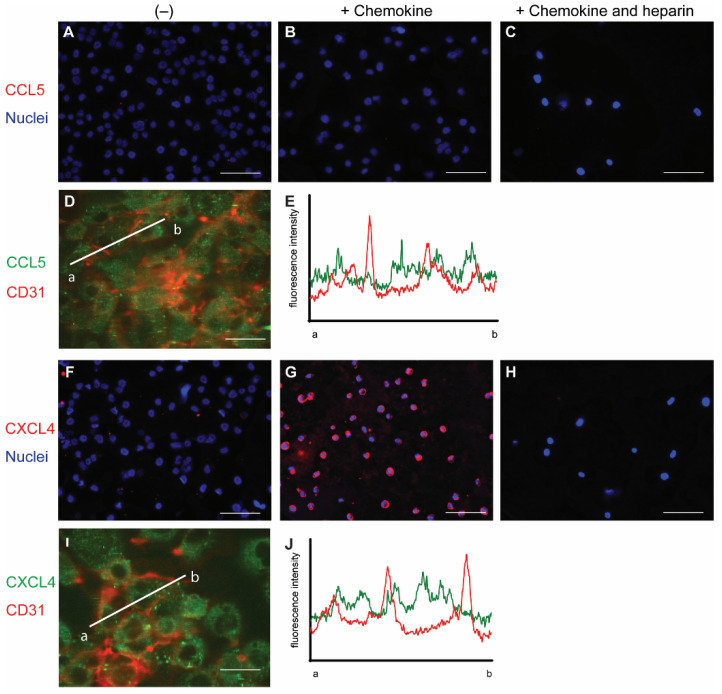
Staining of chemokines after addition to endothelial cells. EAHy were grown on a cell culture slide, mock-treated (**A**,**F**), or incubated with the chemokines CCL5 (**B**–**D**,) or CXCL4 (**G**–**I**) for 60 min at 37 °C, and cells were washed with PBS alone or PBS with 1 mg/mL heparin (**C**,**H**). External chemokines on living cells were then stained with the respective primary antibodies and an Alexa Fluor 647-coupled secondary antibody, and nuclei were visualized with Hoechst 33342. (**A**–**C**,**F**–**H**). Internalized chemokines were stained after fixation and permeabilization of cells using a FITC-coupled secondary antibody (green), and cell membrane was visualized with APC-coupled CD31 (red), using confocal microscopy (**D**,**I**). Intensity profile through adjacent endothelial cells indicated by line a-b in (**D**,**I)** respectively, showing cell membrane (red) and CCL5 (**E**) or CXCL4 (**J**) resp. (green). Scale bar: 100 µm (**A**–**C**,**F**–**H**) or 50 µm (**D**,**I**); (*n* = 4).

**Figure 2 ijms-22-07332-f002:**
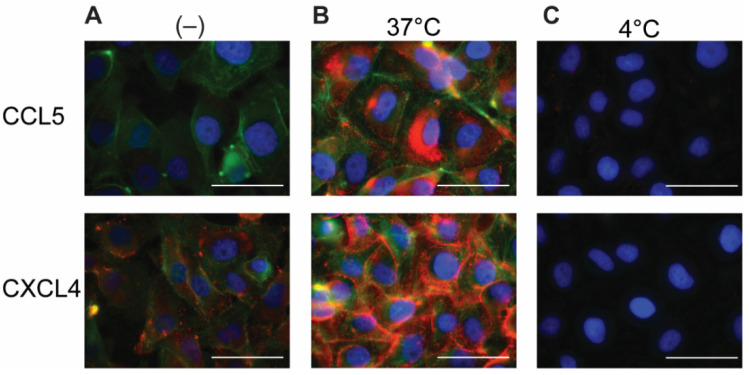
Addition of CCL5 and CXCL4 to EC leads to internalization. EAHy incubated with buffer (**A**) or the chemokines CCL5 (top row) or CXCL4 (bottom row) at 37 or 4 °C (**B**,**C**, respectively) for 60 min and washed with heparin (1 mg/mL) prior to fixation, permeabilization, and staining. CCL5 (red, upper row), CXCL4 (red, lower row), F-actin (green), and nuclei (blue). Scale bar: 50 µm. (*n* = 4).

**Figure 3 ijms-22-07332-f003:**
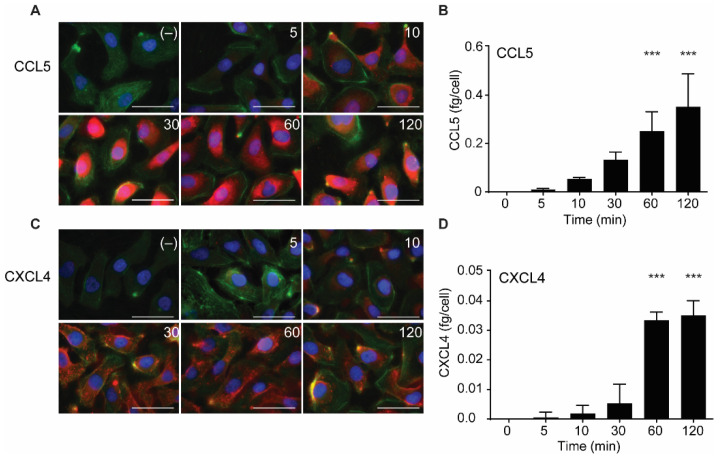
Time-dependent uptake of CCL5 and CXCL4 in EC. CCL5 and CXCL4 were added to EAHy for indicated times at 37 °C. After washing with heparin, cells were fixed, permeabilized, and stained. Fluorescent signals of CCL5 (**A**, red) and CXCL4 (**C**, red), actin (green), and nuclei (blue) (*n* = 3). Quantification of intracellular chemokine concentrations of CCL5 (**B**) and CXCL4 (**D**) by ELISA. *** *p* < 0.001 (*n* = 3), ANOVA with Sidak’s post-test.

**Figure 4 ijms-22-07332-f004:**
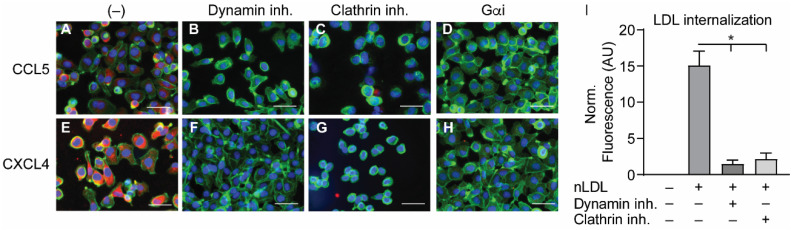
Internalization of CCL5 and CXCL4 is dependent on dynamin- and clathrin-mediated endocytosis, and on G protein-coupled receptor signaling (**A**–**H**). EAHy were pre-incubated without (**A**,**E**) or with inhibitors to dynamin (**B**,**F**), clathrin (**C**,**G**), or GPCR (**D**,**H**) prior to incubation with CCL5 (top row) or CXCL4 (bottom row) at 37 °C. Cells were then washed with heparin prior to fixation and permeabilization and stained for the chemokines (red), F-actin (green), and nuclei (blue). Scale bar: 100 µm. Panel (**I**) shows the internalization of LDL in EAHy, and the blocking thereof by the inhibitors to dynamin or clathrin, as a positive control. * *p* < 0.05, ANOVA with Dunn’s post-hoc test (*n* = 6).

**Figure 5 ijms-22-07332-f005:**
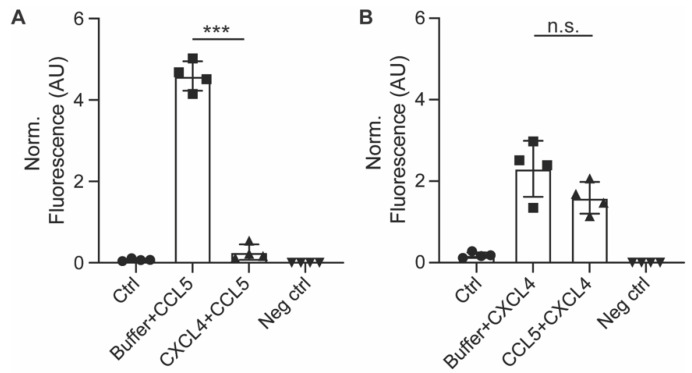
Pre-incubation with CXCL4 inhibits subsequent uptake of CCL5. EAHy cells were pre-incubated with CXCL4 for 30 min prior to incubation with CCL5 for 120 min and its uptake measured by antibody staining after heparin washing (**A**), or vice versa (**B**). *** *p* < 0.001 (*n* = 4), ANOVA with Sidak’s post-test. n.s. = non-significant.

**Figure 6 ijms-22-07332-f006:**
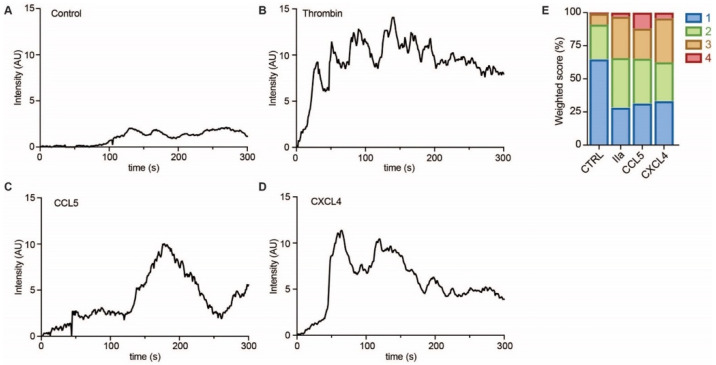
Chemokines induce calcium signaling in EC. HUVECs loaded with Fluo-4 were stimulated with PBS (control, **A**), thrombin (IIa, **B**), CCL5 (**C**), and CXCL4 (**D**), and intensity profiles (in arbitrary units, AU) were recorded of >30 cells in over 3 independent experiments. Representative traces are shown in (**A**–**D**) and traces were classified as described in the methods section and summarized in (**E**).

**Figure 7 ijms-22-07332-f007:**
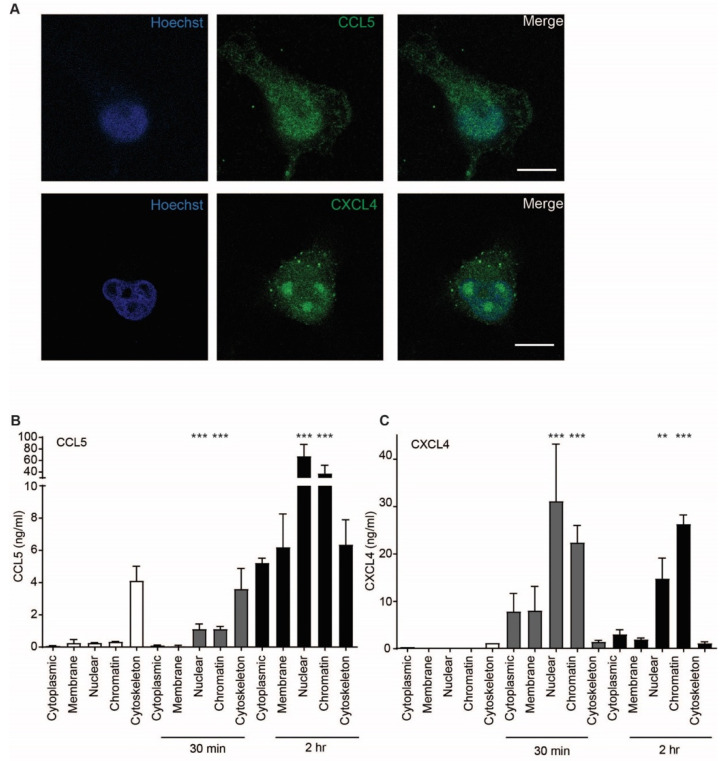
CCL5 and CXCL4 are targeted to the nucleus. (**A**) EAHy were incubated with CCL5 or CXCL4 for 60 min at 37 °C, fixed, and stained for nuclei (blue), chemokine CCL5, and CXCL4 (green). Shown are confocal micrographs. Scalebar = 10 µm. (*n* = 3) (**B**,**C**) EAHy were incubated without (white bars) or with CCL5 (**B**) or CXCL4 (**C**) for 30 or 120 min (gray and black bars, respectively), lysed, and chemokines were determined in subcellular fractions (*n* = 3, ** *p* < 0.01, *** *p* < 0.001, ANOVA with Dunn’s post-test).

**Figure 8 ijms-22-07332-f008:**
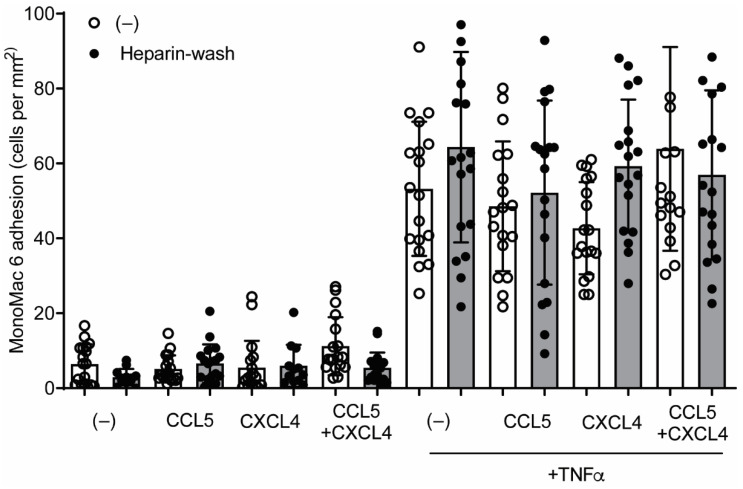
CCL5 and CXCL4 internalization does not affect monocytic cell arrest to endothelial cells. HUVECs were stimulated with 10 ng/mL TNFa for 4 h and/or CCL5 or CXCL4 for 1 h and washed without (black bars) or with heparin (white bars) prior to perfusion of Syto 13-labeled MonoMac-6 cells at 3 dynes/cm^2^. Arrested cells were counted in 6 different fields (*n* = 4).

## Data Availability

The source data presented in this study are openly available at https://surfdrive.surf.nl/files/index.php/s/vOlqyU7Zcv33a9X (accessed on 7 July 2021).

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
