# Peer review of "Rapid Internalization and Nuclear Translocation of CCL5 and CXCL4 in Endothelial Cells"

_ijms, 2021, doi:10.3390/ijms22147332_

Round 1

Reviewer 1 Report

Dickhout et al. investigated fate of CCL5 and CXCL4 after incubating endothelial cells with these chemokines. They report deposition of CXCL4 on endothelial cells and time dependent uptake of CXCL4 and CCL5. In addition, they show chemokine dependent Calcium flux using HUVEC and translocation of chemokines into the nucleus (using EAHy cells). Finally, incubation of endothelial cells with the chemokines did not affect monocyte recruitment in a flow chamber assay. The data presented in this study are potentially interesting and the paper is well written, however I have some concerns regarding the analysis of their findings (especially confocal images are not very convincing).

Major points:

  • The authors need to provide number of performed experiments (n= x biological replicates) for Figure 1, 2, 3A and C, 4, 6A.
  • In addition, they should try to quantify the findings in their images. For example, the authors could analyze and quantify (not only visualize) co-localization of surface bound chemokine with molecules expressed on the cellular surface (e.g. VE-Cadherin/PECAM-1). An image alone (one image each finding) is not convincing and sufficient for the conclusions of surface bound chemokine, intracellular chemokine, and chemokine located within the nucleus (although intracellular and nuclear deposition was underlined with determination of chemokine in subcellular fractions (6B and C) and ELISA (3B and D).
  • Figure 1: The authors claim to investigate and show chemokine binding to an endothelial monolayer. However, they did not stain cell-cell contacts, but only the nucleus. The ability of endothelial cells to take up chemokines may be dependent on the density of the cells (single endothelial cell versus cellular monolayer). Therefore, the authors should provide bright field micrographs or use a surface marker (e.g. VE-Cadherin, PECAM-1). This would also help to visualize and analyze localization of the chemokine on the cellular surface. Figure 1C and 1F: the number of endothelial cells seems very low (hard to believe that these images show a monolayer).
  • Figure 2:The staining in Figure 2 is not very convincing. Only Figure 2A shows a (even very weak) signal for F-actin. Where is the F-actin signal in B and in C? In addition, in Figure 6A the authors argue to see the chemokine within the nucleus after 60min. of incubation time. In Figure 2B the signal is clearly outside the nucleus. What does this mean? How do the authors explain that discrepancy? Did the authors treat the cells in a different way in Figure 2 and 6?
  • Figure 4: Image C and G does not look like an endothelial monolayer, but rather some single cells. VE-Cadherin or PECAM-1 staining would help to see if it is indeed a monolayer.
  • Figure 5: please show a larger field of view, with more cells. Again, this image does not seem to reflect a monolayer staining, but rather a single cell.
  • Supplemental Figure S1A and C: Why do the nuclei in the images (–) not show any chemokine staining (as you would expect from Figure 5)? And why are they positive (I guess this is co-localization of chemokine and Höechst) in (–GAG) and (TAK-779)? Maybe the authors can provide single channel views in addition to merged images and comment on the discrepancy.

Minor points:

  • Figure legend 2: Table 5 should be CCL5
  • Figure legend 4: Scheme 5 should be CCL5
  • Figure 5C: please align x-axis
  • Figure 5/7: why did the authors use HUVEC in this experiment and EAHy in the other experiments?
  • Figure 7: Please introduce PF4 or call the chemokine CXCL4

Author Response

Please see the uploaded pdf file

Reviewer 2 Report

This manuscript describes studies of the internalization and intracellular fates of chemokines CCL5 and CXCL4 in endothelial cells.  These and other related messenger molecules are thought to play roles in communication between sites of infection and circulating populations of leukocytes, mediating their association with the endothelium and transit across the vascular wall.  The authors report that exogenously-applied CCL5 and CXCL4 are internalized into endothelial cells in monolayer culture (HUVEC or EA.hy926 cells) in a time-dependent manner, eventually partitioning into the nuclear subcellular fraction.  Improved characterization of chemokine trafficking among various biological compartments – circulating leukocytes, the vascular lumen, the glycocalyx, the endothelial cell surface, and the tissue interstitium – is crucial for understanding mechanisms of inflammatory processes (both homeostatic and pathologic) as well as responses to infection or neoplasia.  In this context, there is value in elucidating mechanisms of chemokine transit into or through the endothelium.

Comments:

  1. While the authors have demonstrated internalization (temperature- and time-dependent accumulation of heparin wash-resistant immunoreactivity) of both chemokines under study, it is a concern that no evidence is presented that this process is mediated by specific chemokine receptors. For CXCL4, this lack is understandable, since the convincing identification of such a receptor has been elusive.  For CCL5, neither of the two pharmacologic antagonists of CCR5 tested (DAPTA and TAK-779) inhibited internalization of CCL5 immunoreactivity.  The reader is left to wonder whether the internalization of both CCL5 and CXCL4 is being mediated by the basal turnover of plasma membrane, bringing with it any material with non-selective chemical affinity for phospholipids or outward-facing heparinoids.  This type of mechanism, while unspecific, could still be meaningful as a means of endothelial transit.  What are the authors’ options for measuring basal plasma membrane turnover?
  2. The concentration of chemokines used in these studies (500 ng/ml) seems high. Biological responses to CCL5 reported by the vendor top out at 10 ng/ml, or 50-fold below the concentration used here.  What is the dose-response relationship for CCL5 internalization as measured in Fig 2 or for calcium mobilization in Fig 5?  This would be another way to test involvement of a receptor-mediated process.
  3. CXCL4 binding and internalization follows an expected pattern: detection of cell (surface)-associated immunoreactivity after 60 min at 37C, with the gradual accumulation of heparin wash-resistant immunoreactivity over time measured up to 120 min. The findings with CCL5, however, are puzzling: intracellular accumulation increasing out to 120 min, but no surface detection at 60 min.  Does this imply that the surface bound-material is internalized extraordinarily rapidly, or alternatively that the surface binding sites are of very low density?  Can the authors exclude the possibility that binding of CCL5 at the cell surface leaves its immunologic epitope(s) unavailable for detection?
  4. Both CCL5 and CXCL4 are released from activated platelets, and the cells used in these studies were cultured in the presence of bovine serum.  The authors note that there was negligible surface or internalized CCL5 and CXCL4 immunoreactivity in the absence of exogenous chemokines, which is an important control.  However, do the anti-CCL5 and -CXCL4 polyclonals recognize the bovine proteins?  In any case, does the possibility of prior and chronic exposure of the cells to the chemokines of interest confound the authors’ interpretation?
  5. Lines 204 and 298-299: The loss of chemokine internalization at 4C (Fig 2) can be interpreted as reflecting the slowing of plasma membrane dynamics, not as indicative of “…an active and energy-requiring cellular process” at 37C per se, although the internalization mechanisms undoubtedly require chemical energy. Chemokine that is cell-associated and remains surface-bound and not internalized at 4C should be detectable immunologically (see comment 1 above); is CCL5 detectable on intact cells when incubated at 4C, unlike when incubated at 37C as in Fig 1B?
  6. Line 207: extraneous text “or Table 5.”
  7. Figure 3B and 3D: The graphs are described in the legend as showing intracellular chemokine concentrations. Were these in fact calculated with benefit of estimates of intracellular volume?  If not, a more instructive formulation might be to calculate pg/cell (or whatever the appropriate mass unit would be) – the needed parameters are likely in hand already or readily estimated.

Author Response

Please see the uploaded PDF file

Reviewer 3 Report

Dear Editor,

The manuscript by Dickhout et al. reports an active and directed uptake of the chemokines CCL5 and CXCL4, resulting in an accumulation toward the nucleus in endothelial cellular models.

The design of the study and the technical quality of the work look convincing and results can be of general interest. The manuscript is well-written and easy to follow. However, there is a number of major and minor points that would need to be addressed in order to improve the quality of this paper before it can be accepted for publication:

General:

-Defining abbreviation whenever they are firstly introduced and keep using them throughout the GPCR in lines 35 and 70 and a couple of few others.

Major:

-The authors have not conclusively demonstrated that it is clathrin-mediated since the study lacks a positive control (e.g. transferrin assay) to confirm the pathway.

-Imaging was an essential aspect of this manuscript. Author needs to provide more details such as how many FOVs have been taken and what are their measures to minimize biases, and how they have excluded any possible interference from background signals in order to enhance the reproducibility of the presented data. Magnification number should be included for all the figures. But it won’t be enough as it has nothing to do with resolution especially for the purpose of quantitative analyses like in this study. So, author needs to include NA of the utilized lens as well.

-Line 68 in the introduction: authors have mentioned that this has been performed “under static condition”; however, they haven’t touch on the difference between static vs flow nor the documented importance of the latter as it has recently been shown by Santa-Maria et al JCBFM 2021 regarding the role of flow in the expression of glycocalyx. Reference:

https://pubmed.ncbi.nlm.nih.gov/33563079/

-In addition to the above point about static vs flow, Author needs to mention the limitations of this study. For example, the findings from this study are based on 2D cultures. Physiologically, cells never grow in 2D. Co-cultures and so 3D perfused cellular models should provide a more physiologically relevant insight and this might affect the underlying signaling mechanism and function. Authors also need to discuss future directions which can benefit from 3D self-organized models and human brain microvessel-on-a-chip platforms, especially those amenable for advanced imaging to monitor endocytosis in real-time and also TEM. References to be included:

-https://pubmed.ncbi.nlm.nih.gov/25033469/

-https://pubmed.ncbi.nlm.nih.gov/33117784/

- In Figure 7, authors reported that CCL5 and CXCL4 internalization does not affect monocytic cell arrest to endothelial cells. This might be due to the very limited number of samples (n=4) and the huge variations between samples as indicated by the error bars. Authors need to discuss this limitation at the beginning of the discussion and bar charts should be replaced with much more informative scatter plots or similar to enhance the visibility of the data and provide a more transparent insight to the distribution of the presented results. A reference for further details:

https://onlinelibrary.wiley.com/doi/epdf/10.1111/ejn.13400

Minor:

-A statement regarding their measures to investigate mycoplasma contamination needs to be mentioned as this can significantly interfere with the conclusion of this study.

-“Point 2.8: Kitchen et al STAR protocol 2020 has recently reported a detailed method about dynamic Ca imaging and water flux which can be applied for endothelial cells. Authors need to mention this and discuss the potential for it use in future studies. Reference to be included:

https://pubmed.ncbi.nlm.nih.gov/33377051/

Best.

Author Response

Please see the uploaded pdf file

Round 2

Reviewer 3 Report

Dear Editor,

The authors have successfully addressed the majority of my comments and concerns in order to improve the quality of the manuscript.

I believe that the new data, corrections, additional sections and updated references, have contributed to enhancing the clarity of the manuscript, which I can now endorse for publication.

All the best!